# Testing Fourier Sparsity via Implicit Sensing

**Arijit Ghosh**
Indian Statistical Institute
Kolkata 700108, India
`arijitiitkgpster@gmail.com`

**Subhamoy Maitra**
Indian Statistical Institute
Kolkata 700108, India
`maitra.subhamoy@gmail.com`

**Manmatha Roy**
Indian Statistical Institute
Kolkata 700108, India
`reach.manmatha@gmail.com`

## Abstract

Boolean functions constitute a fundamental object of study in machine learning and theoretical computer science. Among their various complexity measures, Fourier sparsity, the number of nonzero coefficients in a function's Fourier expansion, serves as a natural indicator of structural simplicity. For more than three decades, the problem of learning Boolean functions with sparse Fourier representations has occupied a central place in computational learning theory. A major line of progress has produced algorithms whose complexities depend primarily on the sparsity parameter itself. However, these methods typically assume that this parameter is known in advance. In this work, we explore the problem of Fourier sparsity testing, which naturally relates to this question. Given query access to a Boolean function $f : \mathbb{F}_2^n \to \{-1, +1\}$, we seek to determine whether it is $s$-Fourier sparse or far (under Hamming distance) from every such function.

Our contributions are twofold. On the algorithmic side, we design a new tester with query complexity $\widetilde{O}(s^4)$, independent of the ambient dimension. On the lower bound side, we prove that any tester requires at least $\Omega(s)$ queries. Both bounds improve upon the best known results of Gopalan et al. (SICOMP 2011), who obtained a tester with query complexity $\widetilde{O}(s^{14})$ and a lower bound of $\Omega(\sqrt{s})$. For the upper bound, we introduce a refined notion of a sampler inspired by the junta testing framework and combine it with $\ell_1$-minimization-based compressed sensing techniques. In doing so, we develop a novel method for sampling leaves of parity decision trees associated with Fourier-sparse Boolean functions. The lower bound is obtained via a reduction from communication complexity, leveraging structural properties of Fourier coefficients of a specific class of cryptographically hard functions.

## 1 Introduction

Boolean functions are fundamental in machine learning and theoretical computer science, as they naturally model decision rules in binary classification, logical circuits, and related computational processes. They have been extensively studied in learning theory (Kearns & Vazirani, 1994), complexity theory (O'Donnell, 2014), and cryptography (Carlet, 2020). Among various structural measures, one that is particularly important in the literature on learning theory is Fourier sparsity, which counts the number of nonzero coefficients in the Fourier expansion over $\mathbb{F}_2^n$. Formally, for $f : \mathbb{F}_2^n \to \{-1, +1\}$, its Fourier expansion is given by,

$$f(x) = \sum_{\alpha \in \mathbb{F}_2^n} \widehat{f}(\alpha) \chi_\alpha(x), \qquad \chi_\alpha(x) = (-1)^{\langle x, \alpha \rangle},$$

The Fourier support of $f$ is defined as the set of nonzero Fourier coefficients, i.e., $\mathrm{supp}(\widehat{f}) = \{\alpha : \widehat{f}(\alpha) \neq 0\}$. We say that $f$ is $s$-Fourier sparse if $|\mathrm{supp}(\widehat{f})| \leq s$.

Many natural classes of Boolean functions, such as hypergraph cut functions and bounded-depth decision trees, are Fourier sparse, and learning such functions has long been central in computational learning theory. The study of learning Fourier-sparse Boolean functions dates back to the pioneering work of Kushilevitz & Mansour (1993), inspired by Goldreich & Levin (1989). Over the past two decades, there has been renewed interest in exact reconstruction with complexity primarily dependent on the sparsity parameter. Two main algorithmic paradigms are based on the Sparse Hadamard Transform (Stobbe & Krause, 2012; Indyk et al., 2014; Hassanieh et al., 2012) and compressed sensing techniques Haviv & Regev (2017). A common assumption in these lines of work is that the sparsity level is known in advance, which may not hold in many practical settings. Moreover, exact reconstruction methods, such as $\ell_1$-minimization based compressed sensing, succeed only when the function is exactly, or sufficiently close to, $s$-Fourier sparse, typically requiring either exact sparsity or Hamming distance at most $o(1/s)$ from the nearest $s$-sparse function. Thus, some form of certification of Fourier sparsity is important for obtaining meaningful recovery guarantees. We study the problem of testing Fourier sparsity under membership query access, for Boolean functions $f : \mathbb{F}_2^n \to \{-1, +1\}$, where the tester can query $f(x)$ on inputs of its choice but has no access to its internal structure.

> **Problem Statement:** Given a parameter $\epsilon > 0$ and black-box query access to a Boolean function $f : \mathbb{F}_2^n \to \{-1, +1\}$, decide whether $f$ is $s$-Fourier sparse or $\epsilon$-far from every $s$-Fourier sparse Boolean function, where distance is measured under the Hamming sense:
> $$\text{dist}_0(g, h) = \Pr_{x \in \mathbb{F}_2^n} [g(x) \neq h(x)].$$

The efficiency of a Fourier sparsity tester is measured by its query complexity, i.e., the number of queries it makes to the target function.

**Related Work.** A related version has been studied for real-valued functions $f : \mathbb{F}_2^n \to \mathbb{R}$, where distance is measured via the Euclidean norm,
$$\text{dist}_2(f, g) = (\mathbb{E}_x[(f(x) - g(x))^2])^{1/2}.$$

Yaroslavtsev & Zhou (2020) showed that testing closeness to $s$-Fourier-sparse functions under this metric can be achieved with $\widetilde{O}(s)$ queries, and they established an $\Omega(\sqrt{s})$ lower bound. More recently, Ghosh & Roy (2025) presented a simpler tester with $\widetilde{O}(s)$ query complexity together with a matching $\Omega(s)$ lower bound, thereby nearly closing the gap up to polylogarithmic factors. However, one should note that while closeness in the Hamming sense implies closeness in the Euclidean sense, the converse may not be true. To illustrate, consider two scenarios. In the first case, if $f$ is an exactly $k$-Fourier-sparse Boolean function with precisely $k$ nonzero Fourier coefficients, then each of these coefficients must be large in magnitude, i.e., $\Omega(1/k)$. In the second, a Boolean function with $k$ large Fourier coefficients but also a tail of many small, nonzero coefficients with small $\ell_2$ norm. A $k$-Fourier sparsity tester under the Euclidean distance would accept both types of functions, whereas a tester under the Hamming distance should accept only the first type. This makes testing Fourier sparsity in the Hamming sense comparatively harder: one must certify not only the presence of $k$ large Fourier coefficients but also the absence of a Fourier tail.

The problem of testing Fourier sparsity under Hamming distance was first studied by Gopalan et al. (2011), who gave a non-adaptive tester with query complexity $O(s^6 \log s / \epsilon^2 + s^{14} \log s)$ and proved a lower bound of $\Omega(\sqrt{s})$. Furthermore, Fourier sparsity is affine-invariant, in the sense that it remains unchanged under affine transformations of the domain. Consequently, it is, at least in principle, amenable to testing via general regularity-based frameworks(Kaufman & Sudan, 2008; Bhattacharyya et al., 2013; Hatami & Lovett, 2013); however, such approaches incur impractical tower-type query complexities.

**Our Contributions.** We close the gap between existing upper and lower bounds for Fourier sparsity testing. Our main results are as follows.

- **Upper bound.** We design a non-adaptive tester with query complexity $\widetilde{O}(s^4)$, substantially improving upon the previous $\widetilde{O}(s^{14})$ bound of Gopalan et al. (2011).

**Theorem 1.1.** *Let $s > 0$ and $\epsilon > 0$, and let $f : \mathbb{F}_2^n \to \{-1, +1\}$ be an unknown Boolean function accessible via membership queries. There exists a non-adaptive property testing algorithm that distinguishes whether $f$ is $s$-Fourier sparse or $\epsilon$-far (in Hamming distance) from every such function, with success probability at least $2/3$, using $\widetilde{O}\big(\max\{s^2, 1/\epsilon\} \cdot s^2\big)$ queries.*[1]

- **Lower bound.** We prove a quadratically stronger lower bound of $\Omega(s)$, improving upon the previous best $\Omega(\sqrt{s})$ bound of Gopalan et al. (2011).

  **Theorem 1.2.** *Any (possibly adaptive) property testing algorithm that distinguishes whether a Boolean function is $s$-Fourier sparse or $(1/4)$-far (in Hamming distance) from every $s$-Fourier sparse function, with success probability at least $2/3$, must make at least $\Omega(s)$ queries.*

For the upper bound, we design a tester that refines the notion of a sampler from the junta testing framework of Chakraborty et al. (2011) and combines it with $\ell_1$-minimization based compressed sensing framework. A main ingredient of our algorithm is a new sampling method for parity decision trees, which arises naturally in the analysis of Boolean functions. On the other hand, for the lower bound, we achieve a quadratic improvement by reducing Fourier sparsity testing to a certain linear-algebraic problem in communication complexity. Exploiting structural properties of the Maiorana–McFarland family (McFarland (1973)), we show that any tester, adaptive or non-adaptive, must make at least $\Omega(s)$ queries. Our techniques are of independent interest and have potential applications in learning theory, property testing, and other algorithmic questions related to harmonic analysis.

## 2 PRELIMINARIES

We use the following notations and background results in the rest of the paper.

- By Boolean function we mean functions of the form $f : \mathbb{F}_2^n \to \{-1, +1\}$.
- Given $f : \mathbb{F}_2^n \to \mathbb{R}$, the expected value $\mathbb{E}_x[f]$ is defined as $\mathbb{E}_x[f] := 2^{-n} \sum_{x \in \mathbb{F}_2^n} f(x)$.
- For $\alpha = (\alpha_1, \ldots, \alpha_n)$ and $\beta = (\beta_1, \ldots, \beta_n)$ in $\mathbb{F}_2^n$, their inner product is $\langle \alpha, \beta \rangle := \sum_{i=1}^n \alpha_i \beta_i$.
- For $f, g : \mathbb{F}_2^n \to \mathbb{R}$, their inner product is $\langle f, g \rangle = 2^{-n} \sum_{x \in \mathbb{F}_2^n} f(x)g(x)$.
- Given $\alpha \in \mathbb{F}_2^n$, character function $\chi_\alpha : \mathbb{F}_2^n \to \{-1, +1\}$ corresponding to $\alpha$ is defined as $\chi_\alpha(x) := (-1)^{\langle \alpha, x \rangle}$. Note that the character functions $\{\chi_\alpha : \alpha \in \mathbb{F}_2^n\}$ are orthogonal, that is,

$$\langle \chi_\alpha, \chi_\beta \rangle = \begin{cases} 0 & \text{if } \alpha \neq \beta \\ 1 & \text{if } \alpha = \beta, \end{cases}$$

  and character functions also forms a basis for all real-valued functions on $\mathbb{F}_2^n$.

- Fourier transformation of a function $f : \mathbb{F}_2^n \to \mathbb{R}$ is defined as $\widehat{f}(\alpha) := \langle f, \chi_\alpha \rangle = \mathbb{E}_x[f(x)\chi_\alpha(x)]$, for all $\alpha \in \mathbb{F}_2^n$. By, Fourier inversion formula, for all $x \in \mathbb{F}_2^n$, we have

$$f(x) = \sum_{\alpha \in \mathbb{F}_2^n} \widehat{f}(\alpha)\chi_\alpha(x)$$

- For a function $f : \mathbb{F}_2^n \to \mathbb{R}$, Parseval's identity says that $\langle f, f \rangle = \sum_{\alpha \in \mathbb{F}_2^n} \widehat{f}(\alpha)^2$. Additionally, if $f$ is also a Boolean functions then $\langle f, f \rangle = 1$.
- For any two functions $g, h : \mathbb{F}_2^n \to \mathbb{R}$, from Plancherel theorem we have $\langle g, h \rangle = \sum_{\alpha \in \mathbb{F}_2^n} \widehat{g}(\alpha)\widehat{h}(\alpha)$.
- Given Boolean functions $f, g : \mathbb{F}_2^n \to \{-1, +1\}$, the Hamming distance ($\ell_0$-distance) is defined as $\delta(f, g)$ is $\delta(f, g) := \frac{d(f,g)}{2^n}$. where $d(f, g) := |\{x \in \mathbb{F}_2^n : f(x) \neq g(x)\}|$.
- For $A \in \mathbb{F}_2^{m \times n}$, $\mathrm{rank}(A)$ denotes its rank over $\mathbb{F}_2$.

---

[1] Here $\widetilde{O}(\cdot)$ hides factors polynomial in $\log s$.

## 3 IMPROVED UPPER BOUND FOR TESTING FOURIER SPARSITY

### 3.1 PROOF OUTLINE OF THEOREM 1.1

The approach of employing learning algorithms for property testing, introduced by Goldreich et al. (1998), is founded on the principle that any proper learning algorithm for a function class $\mathcal{C}$ can be converted into a property tester for $\mathcal{C}$. Nevertheless, while proper learning for most Boolean function classes requires at least $\Omega(\log n)$ queries, property testing typically targets sublogarithmic or even constant query complexity, independent of the ambient dimension $n$. A significant advancement in this context was made by Diakonikolas et al. (2007), who introduced the paradigm of testing via implicit learning. The key insight underlying this framework is that many natural classes of Boolean functions, such as monotone DNFs, decision lists, decision trees, branching programs, Boolean formulas, sparse polynomials over $\mathbb{F}_2$, and Boolean circuits admit succinct representations and are well-approximated by junta functions. For instance, every $s$-term DNF is $\epsilon$-close to another $s$-term DNF depending on only $O(\log s + \log(1/\epsilon))$ variables. The testing-via-implicit-learning framework (Diakonikolas et al., 2007) exploits structural approximation along with the invariance of junta functions under permutations of input variables. Building on this principle, Chakraborty et al. (2011) proposed a query-efficient sample extractor that, given query access to a function promised to be close to a junta, simulates access to the underlying junta up to a permutation of variables. This approach yields improved and in some cases optimal query bounds for testing various subclasses of juntas.

Returning to the problem of testing Fourier sparsity, we recall that Gopalan et al. (2011) designed their tester by exploiting certain locally testable structural properties of Fourier-sparse functions. In particular, they leveraged the granularity of nonzero Fourier coefficients, namely, that each nonzero coefficient must have sufficiently large magnitude and used a hashing-based technique to efficiently identify these large coefficients. In contrast, our approach is based on the paradigm of testing via implicit learning. However, a central challenge in our setting is that existing techniques for testing membership in subclasses of junta functions are insufficient for handling Fourier-sparse functions. The reason is that the class of Fourier-sparse functions is strictly more general than the class of juntas. For example, a linear function depending on $O(n)$ variables is 1-Fourier sparse, since it has exactly one nonzero Fourier coefficient. Yet such a function is not a junta, as its output depends on a linear number of input variables. Thus, Fourier sparsity does not imply low variable dependence. Another fundamental distinction is structural: Fourier sparsity is invariant under invertible linear transformations of the domain, whereas the junta property is not preserved under such transformations. Consequently, testing Fourier sparsity via the testing-by-implicit-learning paradigm necessitates new techniques that go beyond the existing junta-testing framework. More specifically, given query access to an unknown Fourier-sparse function, we develop a procedure that extracts uniformly distributed samples from a certain low-dimensional Fourier-sparse function. The latter can be viewed as a low-dimensional sketch of the original function, capturing its essential spectral structure in a reduced space.

At the core of our approach is an exact learning procedure for $s$-Fourier sparse Boolean functions (see Algorithm 2). It combines $\ell_1$-minimization techniques from compressed sensing (see Chapter 4 of Moitra (2018)) with the RIP result of Haviv & Regev (2017). The key implication is that any $s$-Fourier sparse function $f : \mathbb{F}_2^n \to \{-1, +1\}$ can be recovered exactly from $O(s \log^2 s \cdot n)$ uniformly random samples by solving an appropriate $\ell_1$-minimization program. However, this sample complexity scales linearly with the ambient dimension $n$, which is prohibitively large for our testing framework. To remove this dependence, we invoke a structural theorem of Sanyal Sanyal (2019), which shows that the linear span of the nonzero Fourier coefficients of an $s$-Fourier sparse function has dimension at most $O(\sqrt{s})$. Consequently, every $s$-Fourier sparse function $f : \mathbb{F}_2^n \to \{-1, +1\}$ admits a low-dimensional representation

$$f = f^* \circ L,$$

where $L : \mathbb{F}_2^n \to \mathbb{F}_2^r$ is a linear transformation and $r = O(\sqrt{s})$. Combining this reduction with the RIP guarantee of Haviv & Regev (2017), we obtain that the reduced function $f^* : \mathbb{F}_2^r \to \{-1, +1\}$ can be reconstructed exactly using $O(s^{3/2} \log^2 s)$ random samples, now independent of $n$. Motivated by this observation, our tester reduces Fourier sparsity testing to reconstructing an appropriately compressed version of the target function. Concretely, we reconstruct a composition of the form $f \circ U \circ R$, where $U : \mathbb{F}_2^n \to \mathbb{F}_2^n$ is an unknown invertible linear transformation and $R : \mathbb{F}_2^n \to \mathbb{F}_2^r$

is a known linear map. The main technical ingredient enabling this reduction is Algorithm 3, a sampler in the spirit of the extractor of Chakraborty et al. (2011) from the junta-testing framework. Our sampler substantially generalizes that approach and generates uniform random samples from the composed function $f \circ U \circ R$ using only a small number of queries to $f$. At its core lies a local list-correction procedure for the Hadamard code, inspired by Datta et al. (2026), who studied the problem of testing linear isomorphism.

## 3.2 PROOF OF THEOREM 1.1

We begin by introducing the notion of coset sampling, which plays a key role in generating samples for the exact learning machinery, albeit in a low-dimensional space. Let $\mathcal{O}_H^f$ denote the sampling procedure with respect to a subspace $H$, given query access to the function $f$, as described in Algorithm 1. Throughout this work, we refer to the samples produced by this procedure as coset samples of $f$ with respect to the subspace $H$.

---

**Algorithm 1: SubspaceExplicitCosetSampler**

---

**Input:** Subspace $H$ with basis $B := \{\beta_1, \beta_2, \cdots \beta_r\}$, and query access to $f$
**Output:** An uniform random sample of the function $f$ restricted to a random coset of $H^\perp$

1. Sample $b$ uniformly at random from $\mathbb{F}_2^r$ and define the coset $C(b)$

$$C(b) = \{\alpha \in \mathbb{F}_2^n \mid \langle \alpha, \beta_i \rangle = b_i\}.$$

2. Select a uniformly random element $p$ from $C(b)$.

3. Return the pair $(b, f(p))$.

---

Let $\mathcal{S}_f$ denote the subspace spanned by the vectors corresponding to the nonzero Fourier coefficients of $f$. Observe that the coset samples generated by $\mathcal{O}_{\mathcal{S}_f}^f$ can be interpreted as uniform samples drawn from the set of leaves of the non-adaptive parity decision tree representation of $f$. For a parameter $\theta > 0$, the $\theta$-restricted Fourier span $\mathcal{S}_f(\theta)$ is defined as the subspace spanned by those vectors $\alpha \in \mathbb{F}_2^n$ for which $|\widehat{f}(\alpha)| \geq \theta$. In this work, we build a query-efficient implementation of $\mathcal{O}_{\mathcal{S}_g(\theta)}^g$, given query access to $f$ and the threshold $\theta > 0$, where $g = f \circ A$ for some unknown invertible linear transformation $A : \mathbb{F}_2^n \to \mathbb{F}_2^n$.

**Lemma 3.1.** *There exists an algorithm **SubspaceImplicitCosetSampler** (Algorithm 3) that given query access to a Boolean function $f : \mathbb{F}_2^n \to \{-1, +1\}$, a threshold $\theta > 0$, and a parameter $\lambda \in \mathbb{N}$, generates a set of $\lambda$-many uniform coset samples $\xi$, with respect to the subspace $\mathcal{S}_{f \circ A}(\theta)$, for a function $f \circ A$ where $A$ is some non-singular (and possibly unknown) linear transformation of $\mathbb{F}_2^n$. The query complexity of the algorithm is $\widetilde{O}\left(\frac{1}{\theta^4} + \max\left\{\frac{1}{\theta^2}, \lambda\right\} \cdot \frac{1}{\theta^2}\right)$, and the failure probability of the algorithm is at most $\frac{1}{10}$.*

While the proof of Lemma 3.1 is deferred to the later part of this section, we assume its validity for now and proceed to demonstrate how it can be used to establish Theorem 1.1.

***Proof of Theorem 1.1***. In Algorithm 2, the subroutine **SubspaceImplicitCosetSampler** (Algorithm 3) is invoked with parameters $\theta$ and $\lambda$. By Lemma 3.1, this procedure produces $\lambda$ uniformly random coset samples (as described in Algorithm 1) with respect to the subspace $\mathcal{S}_{f \circ A}(\theta)$. Using these coset samples, we define a low-dimensional Boolean function $f^* : \mathbb{F}_2^r \to \{-1, 1\}$, where $f^* = f \circ T \circ A$, and $T : \mathbb{F}_2^n \to \mathbb{F}_2^r$ is a linear transformation mapping the standard basis vectors $e_i^n$ of $\mathbb{F}_2^n$ to the corresponding standard basis vectors $e_i^r$ of $\mathbb{F}_2^r$, for $i \in \{1, 2, \ldots, r\}$. The transformation $T$ also maps each of the $2^r$ cosets of $\mathcal{S}_{f \circ A}(\theta)^\perp$ (the orthogonal subspace of $\mathcal{S}_{f \circ A}(\theta)$) to unique elements of $\mathbb{F}_2^r$.

The next step is to determine whether the function $f^*$ is $s$-Fourier sparse. It is kind of folklore (see Chapter 4 of Moitra (2018)) that for any Boolean function $h : \mathbb{F}_2^r \to \{-1, +1\}$ with at most

$s$ nonzero Fourier coefficients, if there exists an $m \times 2^r$ subsampled Walsh–Hadamard matrix $M$ satisfying the Restricted Isometry Property (RIP) of order $k$ with constant $\delta_k < \frac{1}{3}$ (and $\delta_{2k} + \delta_{3k} < 1$), then the Fourier spectrum of $h$ can be recovered exactly with high probability. This recovery is achieved by solving the following $\ell_1$-minimization problem using $m$ uniformly random labeled examples $(x, y = h(x))$:

$$\widehat{h} = \arg\min \|\widehat{h}\|_1 \quad \text{subject to} \quad M\widehat{h} = y.$$

A bound on $m$ determines the number of required random examples and, consequently, the number of coset samples needed from $f \circ A$. To establish this bound, we recall the result of Sanyal (2019), which states that for any Fourier sparse function $f_A$, the dimension of the subspace spanned by its nonzero Fourier coefficients is at most $O(\sqrt{s})$. Following result of Haviv & Regev (2017), says that $m = O\left(s^{3/2} \cdot \text{poly}(\log s)\right)$ random samples suffice for exact recovery of $f^*$.

**Lemma 3.2** (Haviv & Regev (2017)). *Let $M \in \mathbb{C}^{N \times N}$ be a unitary matrix satisfying $\|M\|_\infty \leq O(1/\sqrt{N})$, and let $\delta > 0$ be sufficiently small. Construct a measurement matrix $B \in \mathbb{C}^{q \times N}$ by selecting $q$ rows uniformly and independently from $M$, each scaled by $\sqrt{N/q}$. If*

$$q = O\left(\log^2(1/\delta) \cdot \delta^{-2} \cdot k \cdot \log^2(k/\delta) \cdot \log N\right),$$

*then, with probability at least $1 - 2^{-\Omega(\log N \cdot \log(k/\delta))}$, the matrix $B$ satisfies the Restricted Isometry Property of order $k$ with isometry constant $\delta$.*

---

**Algorithm 2: EffcientFourierSparsityTester**

**Input:** Fourier sparsity $s$, and query access to $f$,
**Output:** Whether $f$ has fourier sparsity $s$ or $\varepsilon$ far from every such function
**Initialization:** $\theta = \frac{1}{4s}$, $\lambda = \max\left\{\frac{1}{\varepsilon}, Cs^2 \cdot \text{poly}(\log s)\right\}$ where $C$ is some large constant

1. $(\zeta, r) \leftarrow$ **SubspaceImplicitCosetSampler**$(\theta, \lambda)$

2. Initialize $\Phi$ such that for each $(x, y) \in \zeta$ and for each $z \in \mathbb{F}_2^r$, set

$$\Phi[x][z] = \frac{2^{\frac{r}{2}}}{\lambda^{\frac{1}{2}}} \cdot (-1)^{x \cdot z}.$$

3. Solve the following optimization problem:

$$\min \sum \|\hat{h}\| \quad \text{subject to} \quad \langle \Phi[x][*] \cdot \hat{h}\rangle = y, \quad \text{for all } (x, y) \in \zeta.$$

4. If $\hat{h}$ corresponds to the Fourier spectrum of some $s$-Fourier sparse Boolean function, accept; otherwise, reject.

---

We now show that if the input function $f$ is $s$-Fourier sparse, then the algorithm always accepts.

**Lemma 3.3** (Completeness). *If the function $f$ is $s$-Fourier sparse, then Algorithm 2 accepts.*

*Proof.* We begin by recalling a result from Gopalan et al. (2011), which states that if a function $f$ is $s$-Fourier sparse, then all of its nonzero Fourier coefficients are integer multiples of $\frac{1}{2^{\lceil \log s \rceil}}$. Moreover, Fourier sparsity is invariant under nonsingular linear transformations. When the subroutine **SubspaceImplicitCosetSampler** (Algorithm 3) is invoked with the threshold parameter $\theta = \frac{1}{4s}$, Lemma 3.1 guarantees that it indeed discovers the Fourier span of $f \circ A$, for some unknown invertible linear transformation $A$. Additionally, the algorithm produces $\lambda$ uniform coset samples, where

$$\lambda = \max\left\{\frac{1}{\varepsilon}, \ O\left(s^2 \cdot \text{poly}(\log s)\right)\right\}.$$

By the guarantees of the sparse recovery algorithm discussed previously, these samples suffice to exactly reconstruct the Fourier spectrum of the projected function $f^* = f \circ T \circ A$. Hence, the algorithm accepts. $\qquad\square$

**Lemma 3.4** (Soundness). *If the function $f$ is $\varepsilon$-far from every $s$-Fourier sparse Boolean function, then Algorithm 2 rejects.*

*Proof.* We argue by contraposition. Suppose Algorithm 2 accepts. Then there exists a function $\widetilde{f}$, obtained via composition with appropriate linear transformations, such that it agrees with $f$ on all $\lambda$ uniform samples collected during Step 4 of Algorithm 3, and moreover $\widetilde{f}$ is $s$-Fourier sparse. Let us define the set **GOOD** as the set of inputs where $\widetilde{f}$ and $f$ agree, and **BAD** as the complement. The probability that all $\lambda$ samples fall within the **GOOD** set is $\left(1 - \frac{|\textbf{BAD}|}{2^n}\right)^\lambda$. If this probability is at least a constant, say $\Omega(1)$, then by a standard union bound and Markov's inequality, we must have $\frac{|\textbf{BAD}|}{2^n} \leq \frac{1}{O(\lambda)}$, which implies that $f$ is $\frac{1}{O(\lambda)}$-close to some $s$-Fourier sparse function $\widetilde{f}$. $\square$

We now analyze the query complexity of Algorithm 2. The algorithm initializes with parameters $\theta = O(1/s)$ and $\lambda = \max\left\{\frac{1}{\varepsilon}, \widetilde{O}(s^2)\right\}$. The only queries made are via calls to Algorithm 3. According to Lemma 3.1, the total number of queries made within that algorithm is bounded by $\widetilde{O}\left(\frac{1}{\theta^4} + \max\left\{\frac{1}{\theta^2}, \lambda\right\} \cdot \frac{1}{\theta^2}\right)$ Substituting $\theta = O(1/s)$, we obtain the final query complexity: $\widetilde{O}\left(s^4 + \max\left\{s^2, \frac{1}{\varepsilon}\right\} \cdot s^2\right)$.

One may verify that Algorithm 2 fails with probability at most $\frac{1}{10} + o(1) < \frac{1}{3}$. $\square$

### 3.3 Proof of Lemma 3.1

We prove this lemma by showing that Algorithm 3 satisfies the theoretical guarantees of Lemma 3.1.

---

**Algorithm 3: SubspaceImplicitCosetSampler**

---

**Input:** Threshold $\theta$, Number of samples $\lambda$, Distance $\epsilon$
**Output:** Coset samples of $f \circ A$ with respect to subspace $\mathcal{S}_{f \circ A}(\theta)$
**Parameters:** $\kappa \leftarrow \max(400\theta^2, \lambda)$, $\gamma \leftarrow \log(100\kappa)$

**Step 1:** Run ECONOMICALSIEVE with parameters $\theta, \lambda$ and oracle access to $F$.
**Step 2:** Relabel the columns of $Q$ as $\{B_1, \ldots, B_k\}$ such that

$$B_1 = e_1^n, \ldots, B_r = e_r^n,$$

and for all $i \in \{r+1, \ldots, k\}$, $B_i$ is a linear combination of $\{B_1, \ldots, B_r\}$.
**Step 3: foreach** $x \in \{x_1, x_2, \ldots, x_\kappa\}$ **do**
    Compute $b \in \mathbb{F}_2^r$ where

$$b_j = \frac{1 - Q[x][B_j]}{2}, \quad \forall j \in [r]$$

    Update $\zeta \leftarrow \zeta \cup \{(b, f(x))\}$
**end**
**Step 4:** Construct matrix $H \in \mathbb{F}_2^{r \times n}$ whose rows are $B_1, \ldots, B_r$.
**Step 5: return** $\{H, \zeta, r\}$

---

We use the following local list correction framework from Datta et al. (2026), which plays crucial role in our Algorithm.

**Lemma 3.5** (Datta et al. (2026)). *There exists an algorithm, Economical Sieve, that takes parameters $\theta$ and $\lambda$ as input, makes $\widetilde{O}\left(\max\left(\frac{1}{\theta^4}, \frac{\lambda}{\theta^2}\right)\right)$ queries to the truth table of a Boolean function $f: \mathbb{F}_2^n \to \{-1, 1\}$, and, with probability at least $\frac{9}{10}$, outputs:*

- *A matrix $Q \in \{-1, +1\}^{\lambda \times k}$, where the $(i, j)$-th entry is $\chi_{\alpha_j}(x_i)$;*

- *A column vector $F \in \{-1, 1\}^\lambda$, where $F(i) = f(x_i)$ for each $i \in [\lambda]$,*

*where each $x_i \in \mathbb{F}_2^n$ is drawn independently and uniformly at random, and each $\alpha_j \in \mathbb{F}_2^n$ belongs to a set $\mathcal{S} = \{\alpha_1, \ldots, \alpha_k\}$ satisfying:*

- *For every $\alpha \in \mathbb{F}_2^n$ such that $|\widehat{f}(\alpha)| \geq \theta$, we have $\alpha \in \mathcal{S}$;*

- *For all $\alpha \in \mathcal{S}$, it holds that $|\widehat{f}(\alpha)| \geq \theta/2$.*

In this work, we assume Lemma 3.5 and proceed with the proof of Lemma 3.1. The proof of Lemma 3.5 can be found in Datta et al. (2026).

*Proof.* Observe that the matrix $Q$ yields the heavy Fourier coefficients of $f$ evaluated at uniformly random points $x$. These coefficients, however, are not available in explicit form; rather, they are represented implicitly as dominant elements within their respective cosets. Nevertheless, evaluations of the corresponding characters at uniformly random points enable us to uncover the linear relationships among them.

To see why, let $\mathcal{T}$ be a subset of $\mathcal{S}$, the set of Fourier Coefficients corresponding to columns of $Q$, such that $\sum_{\alpha_i \in \mathcal{T}} \alpha_i = \mathbf{0}^n$. Then for all $x \in \mathbb{F}_2^n$,

$$\prod_{\alpha_i \in \mathcal{T}} \chi_{\alpha_i}(x) = \prod_{\alpha_i \in \mathcal{T}} (-1)^{\langle \alpha_i, x \rangle} = (-1)^{\left\langle \sum_{\alpha_i \in \mathcal{T}} \alpha_i, x \right\rangle} = 1.$$

Thus, the product of the corresponding columns of $Q$, denoted $\Pi_{\mathcal{T}}$, always equals $\mathbf{1}^\lambda$. On the other hand, if $\mathcal{B} \subseteq \mathcal{S}$ consists of linearly independent vectors, then the probability that $\prod_{\alpha_i \in \mathcal{B}} \chi_{\alpha_i}(x) = 1$ for all $x$ is small.

**Claim 3.6.** *The probability that all entries of $\prod_{\alpha_i \in \mathcal{B}} \chi_{\alpha_i}(x_j)$ are equal to 1 is at most $2^{-\lambda}$. Moreover, for any fixed $\mathcal{B}$, the probability that this product equals 1 for any subset of $\mathcal{B}$ is at most $1/100$.*

*Proof.* Since the vectors $\alpha_i$ are linearly independent and the points $x_1, \ldots, x_\lambda$ are sampled uniformly at random from $\mathbb{F}_2^n$, we have $\Pr[\prod_{\alpha_i \in \mathcal{B}} \chi_{\alpha_i}(x_j) = 1] = 1/2$ for each independent sample $x_j$. Hence, the probability that this holds across all $\lambda$ independent trials is at most $2^{-\lambda}$. Furthermore, the number of possible subsets of $\mathcal{B}$ is at most $2^{|\mathcal{B}|} = 2^{O(1/\theta^2)}$. By applying the union bound, the probability that $\prod_{\alpha_i \in \mathcal{D}} \chi_{\alpha_i}(x_j) = 1$ occurs for any subset $\mathcal{D} \subseteq \mathcal{B}$ is at most $2^{-\lambda} \cdot 2^{O(1/\theta^2)} < o(1)$, provided that $\lambda = \Omega((1/\theta^2) \log(1/\theta))$. $\square$

Using this insight, in **Step 2** of the algorithm we extract a basis of size $r$ for the span of the heavy Fourier coefficients. Without loss of generality, after an appropriate change of coordinates, we relabel these basis vectors as the first $r$ standard basis vectors $e_1^n, \ldots, e_r^n$ in $\mathbb{F}_2^n$. Let $H$ denote the subspace spanned by these vectors. We define a matrix $M \in \mathbb{F}_2^{r \times n}$ whose rows are precisely $e_1^n, \ldots, e_r^n$. For each $x \in \{x_1, \ldots, x_\kappa\}$, the coset of $H^\perp$ containing $x$ is uniquely determined by the linear constraint $Mx = b$, where $b \in \mathbb{F}_2^r$ is computed in **Step 3** of the algorithm. Finally, observe that if $x$ is chosen uniformly at random from $\mathbb{F}_2^n$, then the induced distribution over the cosets of $H^\perp$, given by the mapping $x \mapsto Mx$, is uniform over $\mathbb{F}_2^r$. Consequently, each coset of $H^\perp$ is sampled with equal probability.

**Lemma 3.7.** *Given any $G \subseteq \mathbb{F}_2^n$, the distribution of cosets induced by uniform random selection of $x \in \mathbb{F}_2^n$ is uniform over the cosets of the subspace $G$.*

*Proof.* Let $G \subseteq \mathbb{F}_2^n$ be a subspace of dimension $k$. The cosets of $G$ in $\mathbb{F}_2^n$ are of the form $C_z = z + G$ for $z \in \mathbb{F}_2^n$, and there are exactly $2^{n-k}$ distinct cosets. Each coset has size $|C_z| = 2^k$, while $|\mathbb{F}_2^n| = 2^n$. Now choose $x \in \mathbb{F}_2^n$ uniformly at random and consider the coset containing $x$. For any fixed coset $C_z$, the probability that $x \in C_z$ equals $|C_z|/|\mathbb{F}_2^n| = 2^k/2^n = 1/2^{n-k}$. This value is independent of $z$. Hence, every coset is selected with the same probability, and therefore the induced distribution over the cosets of $G$ is uniform. $\square$

By the theoretical guarantees of the Economical Sieve, the query complexity of the algorithm is $\widetilde{O}\left(\frac{1}{\theta^4} + \max\left\{\frac{1}{\theta^2}, \lambda\right\} \cdot \frac{1}{\theta^2}\right)$. Consequently, the overall failure probability is bounded by $\frac{1}{20} + o(1) < \frac{1}{10}$.

$\square$

## 4 IMPROVED LOWER BOUND FOR TESTING FOURIER SPARSITY

We begin by defining a class of Boolean functions known as the Maiorana–McFarland functions. These functions have a long history in theoretical computer science, including applications in circuit lower bounds and studies of Boolean function structure (Paul, 1977; Blum, 1984; Nisan & Szegedy, 1992; Sanyal, 2019). They are also widely used in symmetric-key cryptography, particularly in stream cipher design, where they provide desirable Fourier and autocorrelation properties (Sarkar & Maitra, 2000).

**Definition 4.1.** *Given positive integers $n$ and $r$ with $r \leq n$, the family of Maiorana–McFarland functions, denoted $\mathrm{MM}_{r,n}$ (McFarland, 1973), consists of functions $f : \mathbb{F}_2^n \to \{-1, +1\}$ of the form*

$$g(x, y) = (-1)^{\langle x, \varphi(y) \rangle}, \quad (x, y) \in \mathbb{F}_2^r \times \mathbb{F}_2^{n-r},$$

*where $\varphi : \mathbb{F}_2^{n-r} \to \mathbb{F}_2^r$ is an arbitrary mapping.*

A key property of these functions is that, when composed with linear transformations, their Fourier sparsity is governed by the rank of the transformation. It was proved in Datta et al. (2026). We refer the reader to Datta et al. (2026) for the full argument; in this work, we assume this result as a starting point for our proof.

**Lemma 4.2.** *Let $n = r + \log r$, let $\varphi : \mathbb{F}_2^{n-r} \to \mathbb{F}_2^r$ have $r$ linearly independent outputs, and let $L \in \mathbb{F}_2^{r \times r}$ be linear. Define*

$$g_L(x, y) = (-1)^{\langle Lx, \varphi(y) \rangle}.$$

*Then the Fourier sparsity of $g_L$ satisfies $|\mathrm{supp}(\widehat{g_L})| \leq \mathrm{rank}(L) \cdot r$.*

Our proof proceeds via a reduction from randomized communication complexity. We briefly recall the relevant notions. Randomized communication complexity studies the minimum number of bits that two parties must exchange in order to compute a function, using either shared randomness, while producing the correct output with high probability. In this work, we focus on the following problem. Alice and Bob are given matrices $A, B \in \mathbb{F}_2^{r \times r}$, respectively, and are promised that $\mathrm{rank}(A + B) \in \{r, r/4\}$. Their goal is to determine which of the two cases holds while minimizing the amount of communication, using public randomness. We denote by $R_{1/3}^*(\mathrm{RANK}_{r,r/4})$ the public-coin randomized communication complexity of this problem with error probability at most $1/3$.

**Theorem 4.3** (Sherstov & Storozhenko (2024))**.**

$$R_{1/3}^*(\mathrm{RANK}_{r,r/4}) = \Omega(r^2).$$

Equipped with these definitions, we now establish a Fourier sparsity testing lower bound that is quadratically stronger than the previously known result (Gopalan et al., 2011).

### 4.1 PROOF OF THEOREM 1.2

We prove the theorem via a reduction from the Approximate Matrix Rank problem. Alice has $A$, Bob has $B$, and $C = A + B$ satisfies $\mathrm{rank}(C) \in \{r, r/4\}$. Alice constructs $f_A$, Bob constructs $f_B$, and together they define $f = f_C = f_{A+B}$. By Lemma 4.2, if $\mathrm{rank}(C) = r$, then $|\mathrm{supp}(\widehat{f})| = r^2$, and if $\mathrm{rank}(C) = r/4$, then $|\mathrm{supp}(\widehat{f})| \leq r^2/4$. The following lemma from Datta et al. (2026) quantifies the distance between the two cases; we refer to that work for the full proof.

**Lemma 4.4** (Datta et al. (2026))**.** *If $\mathrm{rank}(C) = r$, then $f$ is at least $(1/4)$-far from any function with Fourier sparsity at most $r^2/4$.*

Using this result, we complete the reduction. Suppose there exists a tester $\mathbb{T}$ for Fourier sparsity with parameter $s$ and query complexity $q(s, 1/4)$, where $s = r^2/4$. We show how such a tester yields a communication protocol for the approximate rank problem.

To simulate a query at a point $x$, Alice evaluates $f_A(x)$ using her input matrix $A$, and Bob evaluates $f_B(x)$ using his input matrix $B$. They then exchange their one-bit outputs. Since the target function satisfies $f_C(x) = f_A(x)f_B(x)$, both parties can compute $f_C(x)$ after exchanging these two bits. Thus, each simulated query requires exactly two bits of communication. Consequently, the total communication cost of the induced protocol is $2\,q(s, 1/4)$ bits. However, by Theorem 4.3, any public-coin protocol that solves the approximate rank problem with constant error requires $\Omega(r^2)$ bits of communication. Therefore, $2\,q(s, 1/4) = \Omega(r^2)$, which implies $q(s, 1/4) = \Omega(r^2)$. Since $s = r^2/4$, we obtain $q(s, 1/4) = \Omega(s)$, establishing the claimed lower bound.

## 5    CONCLUSION

Our algorithm can be adapted to obtain a tolerant tester as well. The key observation is that the subroutine SUBSETIMPLICITCOSETSAMPLER draws $\widetilde{O}(s^2)$ uniform random examples and checks whether there exists an $s$-Fourier-sparse Boolean function supported on a subspace of dimension about $\widetilde{O}(\sqrt{s})$ that agrees with all sampled points. This step can be made robust to achieve tolerance. Specifically, in distinguishing whether a function is $\varepsilon$-close to some $s$-sparse function or $2\varepsilon$-far from every such function, we may allow up to an $\varepsilon$-fraction of the sampled labels to be flipped, and run the reconstruction procedure repeatedly on these perturbed batches of samples. With a mild degradation in complexity parameters, this yields an $\varepsilon$-versus-$2\varepsilon$ tolerant tester.

However, in this work, we focus on the exact (non-tolerant) setting. More broadly, when a tester is used as a preprocessing step for exact learning of Fourier-sparse Boolean functions, it naturally corresponds to a non-tolerant formulation: exact recovery is meaningful only when the target function itself (or is extremely close to) an $s$-sparse Boolean function. If the function is far from this class in Hamming distance, an exactly sparse representation cannot be reliably recovered. In contrast, PAC learning of Fourier-sparse Boolean functions operates under Euclidean ($\ell_2$) notion of distance and in a tolerant regime. It suffices that most of the $\ell_2$ Fourier weight is concentrated on a few coefficients, and the objective is merely to output a hypothesis with small additive $\ell_2$ error (and hence small prediction error), rather than to recover an exactly sparse representation.

## ACKNOWLEDGMENT

Arijit Ghosh acknowledges partial support from the Science and Engineering Research Board (SERB), Government of India, through the MATRICS grant MTR/2023/001527, and from the Department of Science and Technology (DST), Government of India, through grant TPN-104427. Subhamoy Maitra and Manmatha Roy acknowledges support from the ISEA Phase-III initiative of the Ministry of Electronics and Information Technology (MeitY), Govt of India, through Grant No L-14017/1/2022-HRD.

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
