# OpenReview forum: "Testing Fourier Sparsity via Implicit Sensing"
_ICLR.cc/2026/Conference — ICLR 2026 Poster_

### Official Review · Reviewer_jtCc · 2025-10-30

**Soundness:** 3
**Presentation:** 2
**Contribution:** 3
**Rating:** 4
**Confidence:** 3

**Summary:**

This paper studies the property testing problem of Fourier sparsity under Hamming distance. The goal is to decide, given query access to $f :\mathbb{F}^n_2 \rightarrow \{ \pm1 \}$, whether $f$ is $s$-Fourier-sparse or $\epsilon$-far from every such function, with query complexity independent of $n$. The authors give a non-adaptive tester with complexity $\tilde{O}(s^2 \cdot \max(s^2, 1/\epsilon))$, improving the prior bound of $\tilde{O}(s^{14})$. In addition, this paper shows an $\Omega(s)$ lower bound , improving the prior $\Omega(\sqrt{s})$.

For the upper bound, the tester combines a "implicit sensing" sampler (built in the spirit of junta testing) with an $\ell_1$-minimization step over a low-dimensional projection of $f$. The sampler produces uniform coset samples corresponding to leaves of an appropriate parity decision tree; from these, the analysis identifies the span of heavy Fourier characters (up to a linear change of variables), and reduces the task to recovering a sparse spectrum in dimension $O(\sqrt{s})$. This is done via compressed sensing on random Walsh–Hadamard rows. For the lower bound, the authors reduce testing to a linear-algebraic communication problem instantiated with Maiorana–McFarland functions.

**Strengths:**

- The quantitative improvement is significant: although the new upper bound $\tilde{O}(s^4)$ is still a large polynomial, comparing with the previous bound of $O(s^{14})$, cutting the exponent from 14 to 4 indeed materially advances the state of the art.

- The recovery step leverages robust, well-understood RIP machinery for subsampled Hadamard matrices. I appreciate that the paper plugs into a mature toolbox, which increases credibility and portability. The pipeline of (1) isolating a low-dimensional structure via coset sampling, then (2) running a standard compressed-sensing recovery in that reduced space feels neat and intuitive. Conceptually, the RIP lens clarifies why the query complexity loses all dependence on $n$: after the dimension to $O(\sqrt{n})$, the complexity of compressed sensing scales with the sparsity and the log of the reduced dimension, hence becomes independent of the ambient $n$. This aligns with the property-testing goal and makes the result feel principled rather than ad hoc.

**Weaknesses:**

I think the presentation of the paper could be significantly improved. The paper is understandably notation heavy, but this is exacerbated by the fact that a major part of both the algorithm and the upper-bound proof is deferred to the appendix, which makes cross-referencing difficult. It seems the authors intend to use Algorithm 1 as an "oracle" interface for Algorithm 3 to aid exposition; however, Algorithms 1 and 3 have quite different specifications (inputs/outputs, etc.), which limits the usefulness of this design. I suspect this formatting is driven by ICLR’s tight page limits. Rather than splitting essential descriptions and proofs between the main body and the appendix, it might be better to submit to a TCS venue with a more permissive page limit so the proof can be presented in a self-contained manner. I have not checked all proofs in detail, but even at a semantic level there appear to be some inconsistencies:

1. Lemma 4.2 gives only an upper bound on the support size, whereas the proof of Lemma 4.4 refers to it giving an exact equivalence (i.e., both upper and lower bounds).

2. In Algorithm 3, Step 6 constructs b from the signs $Q[x][B_j]$, thus $b$ seems to correspond to $Ux$, where $U$’s rows are the (unknown) $\alpha$'s. However, on Lines 918-933 it is asserted that $Mx = b$ with $M$’s rows being the standard unit vectors. I am not sure how these two statements are reconciled?

3. In the proof of Lemma A.5, when bounding the expectation $E[|\mathcal{P}_C(z)  - \mathcal{P}_C^*(z) |]$, why is it legitimate to drop the absolute value inside the expectation?

**Questions:**

See the above section.

---

> ### Author Response · Authors · 2025-11-21
>
> We are very grateful to the reviewer for pointing out this important gap in the presentation. Indeed, the sparsity tester relies on the \textsc{SubspaceImplicitCosetSampler} routine; while this routine is fully formalized in the appendix, the main paper currently presents only the \textsc{SubspaceExplicitCosetSampler}, which was intended merely to convey intuition for the implicit version. We acknowledge that this mismatch may cause inconvenience for readers. To address this, we will add an additional page to the main body of the paper in order to introduce the \textsc{SubspaceImplicitCosetSampler} directly. We will provide an intuitive explanation of the algorithm in the main text, while deferring the full technical analysis to the appendix.
>
> Furthermore, to improve readability for a broader audience, we will include an intuitive proof sketch for the key lemma (with the formal proof retained in the appendix). We believe that separating the high-level algorithmic ideas into the main paper and placing the detailed analysis in the appendix will make the exposition substantially more accessible. We are also open to any further suggestions from the reviewer to enhance clarity and readability.
>
>
> We now address the inconsistency issues raised by the reviewer.
>
> 1. Issue with Lemma 4.2:  The definition of Fourier sparsity is inherently an upper-bound notion: a function with Fourier sparsity $1$ is also $k$-Fourier-sparse for every $k>1$. For the lower-bound argument, what we require is the existence of two Boolean functions whose Fourier sparsities differ by a multiplicative factor and yet differ on a constant fraction of inputs. In the proof of Lemma~4.4 we show that even when the Fourier sparsity matches the upper bound exactly, the function is already far; if the sparsity is smaller, the Hamming distance can only be larger. We will revise this discussion to make the intended meaning clearer.
>
> 2. $Q$ in the algorithm and $M$ in the analysis: In the analysis we implicitly assume that relabeling has taken place and that the algorithm has already fixed a basis for the spanning subspace. We then relabel the basis using unit vectors, which leads to the specific matrix representation used in the proof. To avoid confusion, we will revise this part of the exposition so that the transition from the algorithm to the analysis is explicit and consistent.
>
> 3. Omission of the modulus operator inside the expectation: In the first case, the expectation is $0$, so including or omitting the modulus does not affect the argument. In the second case, we explicitly use the modulus when upper bounding the expectation. Nevertheless, we agree with the reviewer that adding the modulus operator throughout would improve consistency and clarity, and we will incorporate this change.
>
>
> Finally, we will thoroughly revise the manuscript to correct all typographical errors and remaining inconsistencies.

---

> > ### Author Response · Authors · 2025-11-27
> >
> > Dear Reviewer,
> >
> > We hope you have had the chance to go through our responses. If you have any further questions or require additional clarification, please let us know. we would be happy to address them to the best of our ability.
> >
> > Thank you once again for reviewing our work.

---

### Official Review · Reviewer_xU3b · 2025-10-31

**Soundness:** 4
**Presentation:** 2
**Contribution:** 3
**Rating:** 6
**Confidence:** 4

**Summary:**

This paper studies property testing of the Fourier sparsity of a Boolean function. Specifically, for a parameter s, we want to perform few queries and distinguish between the cases where the Fourier sparsity is at most s, or the function is eps-far in Hamming distance fomr any function of Fourier sparsity at most s. This is a standard setup in property testing, and the problem here is particularly important: many learning/testing/etc algorithms in the literature assume one knows the Fourier sparsity of the function, so to apply these to functions where the Fourier sparsity is unknown and only query access is available, this type of algorithm is needed.

**Strengths:**

- This is a very natural problem in computational learning theory. The setup and sdetting are very clean and general.
- The paper improves the state-of-the-art for both upper and lower bounds. It's nice that the new insights here are able to make progress in both directions.
- The paper is written well and easy to follow for the most part.

**Weaknesses:**

The main weakness is just that the results seem somewhat incremental. I also think the intro/abstract overstate how important the results are; see questions below.

**Questions:**

- It seems misleading to say there's a "gap in the literature" (in the abstract) or a "critical gap" (page 2) about assuming you know the Fourier sparsity when Gopalan et al. already studied this question, and here the bounds are being improved but not completely new. Could you comment on this; is there some aspect that you address that was missing in the prior work?
- I was expecting the problem to test whether the Fourier sparsity is at most s or at least s + g for some parameter g, rather than using Hamming distance of f to an s-Fourier-sparse function. Wouldn't this more accurately test for whether the function can be used in learning algorithms that need small Fourier sparsity?
- Could you highlight which of the algorithmic techniques are most novel/intereting in this context?
- In applications to learning (i.e., other learning algorithms where it is assumed you know the Fourier sparsity), I would imagine that a powering approach might work where you guess the Fourier sparsity is 1, then 2, then 4, then 8, and so on, until the algorithm succeeds, and only lose a poly factor. Are there any important applications where this type of approach doesn't work, and testing the Fourier sparsity like this is important?

---

> ### Author Response · Authors · 2025-11-21
>
> We thank the reviewer for their thoughtful and careful evaluation of our work.
>
> 1. Regarding "Gap in the literature": First, we want to say that much of the work on sparse Fourier transform algorithms and compressed sensing based on subsampled Hadamard matrices gained significant momentum during the 2010-20 period, whereas the conference version of the work of Gopalan et al. appeared in 2009. It is therefore quite natural that the applicability of Fourier-sparsity testers as a preprocessing step for learning algorithms is not discussed in their work. Although several papers in Property Testing literature explicitly observed that property testing can serve as a preprocessing step for learning, this connection is not explicitly spelled out in Gopalan et al., and we believe the reason is simply that research on learning Fourier-sparse functions became a prominent topic briefly after their work appeared. Finally, we would like to clarify that the main contribution of our paper is not to identify this gap, which, in our view, is a natural motivation for studying the problem, but rather to advance the state of the art along this line of research by improving the gap in the exponent in the query complexity from $28$ to $4$. We hope this clarifies our perspective. We will also revise the manuscript to avoid any confusion regarding the contributions.
>
> 2. On Problem definition: While the question of distinguishing functions with Fourier sparsity at most $s$ from those with sparsity at least $s+g$ may appear natural, it does not capture the operational notion of similarity between Boolean functions that is relevant for property testing. The issue is that Fourier sparsity can change drastically even when two Boolean functions are almost identical. For example, let $f$ be the constant function $f(x)=+1$, which has Fourier sparsity $1$. Let $g$ be the function that equals $+1$ everywhere except at a single point $z$ where $g(z)=-1$. It is well known that such a $g$ has Fourier sparsity $2^{n}$, yet $f$ and $g$ differ on only one input, so their Hamming distance is $2^{-n}$. From a testing perspective, $f$ and $g$ are essentially indistinguishable: detecting the exceptional point requires $\Omega(2^{n})$ queries. In contrast, their Fourier sparsity differ by an exponential factor. This illustrates that Fourier sparsity alone is too fragile to serve as a robust property for testing. For this reason, we adopt the problem formulation introduced by Gopalan et al.
>
> 3. Novel Algorithmic Techniques: Firstly our tester introduces a new sampling primitive (Algorithm~3) that  implicitly simulates samples from a low-dimensional projection $f \circ A$ without knowing the linear map $A$. Our method extends the sample-extractor ideas of  Chakraborty--García-Soriano--Matsliah (2011) beyond juntas to the much richer class of Fourier-sparse functions (which may depend on $\Theta(n)$ variables). The key novelty is a two-level concentration analysis over random cosets: an $\ell_2$-type concentration lemma that identifies cosets containing a heavy coefficient, and an $\ell_1$-type concentration lemma that enables accurate estimation of the heavy coefficient itself. This allows us to recover the Fourier span (upto linear transformation) using only $\tilde O(s^2)$ queries. Secondly our tester is the first combine the low dimensionality of $s$-sparse Boolean function with the RIP guarantees of subsampled Hadamard matrices. This yields exact $\ell_1$-recovery of the sparse Fourier vector from $O(s^{3/2})$ random samples in the reduced dimension.
>
> 4. Why not using a learner directly, with increasing values of $s$:  In principle, one could indeed use the learning routine itself in the manner suggested, but this would introduce an additional multiplicative $\log s$ overhead. In our setting, this means that the overall cost would be the complexity of the exact learning algorithm plus that of the sparsity tester. We would like to emphasize that many exact learning routines for Fourier-sparse Boolean functions are already quite expensive. For instance, $\ell_1$-minimization–based compressed sensing methods are not known to run in time polynomial in $n$ and $s$, and can be prohibitively costly in high dimensions. In such situations, a tester like ours is particularly valuable: it allows one to efficiently verify exact sparsity before invoking any computationally heavy reconstruction procedure, thereby avoiding the need to run an expensive learner on functions that are not exactly $s$-sparse. We also note that many Fourier-sparse learning algorithms are not robust in the sense that they guarantee recovery of an $s$-sparse representation only when the target function is exactly $s$-sparse; outside this regime, they provide no meaningful guarantee. For these reasons, an explicit and efficient sparsity tester is, in our view, a useful contribution to this line of research.

---

> > ### Author Response · Authors · 2025-11-22
> >
> > Here is one more clarification to add in connection with Point 2 (problem definition) from our earlier response.
> >
> > One way to interpret our lower-bound argument is as follows: the task of distinguishing between functions of Fourier sparsity $s$ and $s + 3s = 4s$ requires $\Omega(s)$ queries. The specific constant $4$ is not essential, any fixed constant factor would work. In other words, our lower bound shows that approximating the Fourier sparsity of a Boolean function, even within a constant-factor approximation, necessarily requires a number of queries linear in the sparsity parameter.

---

### Official Review · Reviewer_x8Ha · 2025-11-01

**Soundness:** 3
**Presentation:** 2
**Contribution:** 1
**Rating:** 2
**Confidence:** 2

**Summary:**

The paper proposes a tester for determining whether a given Boolean function is $s$-Fourier sparse or $\epsilon$-far from every $s$-Fourier sparse function, using only $O(s^4)$ queries (ignoring polylogarithmic factors) under the Hamming distance. A Boolean function is $s$-Fourier sparse if it has at most $s$ nonzero Fourier coefficients. Furthermore, the paper complements this result by providing an $\Omega(s)$ lower bound on the query complexity of the problem. The authors claim that both these results improve upon the previously best-known bounds of $O(s^{14})$ and $\Omega(\sqrt{s})$, respectively.

Unfortunately, due to time constraints and limited expertise in this specific subarea, I am unable to verify the technical details.

**Strengths:**

The results appear to significantly advance the state-of-the-art bounds for the problem (although I have some strong reservations about this claim; see Weakness). The techniques also seem novel and potentially interesting.

**Weaknesses:**

Theoretically, the upper bound result is very interesting given the claimed improvement over existing bounds. However, I am not convinced that the improvement is correctly compared to the existing results. This is partly because the paper does a poor job at comparing their work with prior works.
For instance, the paper refers to the work of Yaroslavtsev–Zhou (SOSA’20), who study the same problem under the $\ell_2^2$ distance instead of the Hamming ($\ell_1$) distance considered here, and propose a tester using only $O(s)$ queries. However, it seems to me that their result can be extended to the $\ell_1$ setting with only constant-factor loss in accuracy. Indeed, for any two Boolean functions $f, g$, we have
$||f-g||_2^2 = E_x[(f(x)-g(x))^2] = 4 \cdot Pr_x[f(x)\neq g(x)] = 4 \cdot ||f-g||_1$

where $||f - g||_1$ denotes the Hamming distance between $f$ and $g$. Let $F_s$ be the set of s-Fourier sparse functions.

Therefore, the learner of Yaroslavtsev–Zhou, which estimates $||f - F_s||_2^2$ with additive error $\delta$ using $O(s)$ queries, can be used to distinguish whether the Hamming distance of $f$ to the set $F_s$, of $s$-Fourier sparse functions is at most $\delta$ or at least $4\epsilon - \delta$,  ($\delta$ set to $\epsilon / 1000$ to create a suitable gap). This formulation exactly matches the testing problem studied in the present paper. Moreover, the prior tester achieves this with only $O(s)$ queries, whereas the current paper’s tester requires $O(s^4)$ queries (ignoring polylogarithmic factors). Thus, unless I am missing a critical technical distinction, I do not see how the upper bound in this paper improves upon existing results.

**Questions:**

see above (weakness).

---

> ### Author Response · Authors · 2025-11-21
>
> We thank the reviewer for raising the comparison to Yaroslavtsev--Zhou (SOSA'20). The identity
> $$
> \|f-g\|_2^2 = 4 \cdot \delta(f,g)
> $$
> holds only when both $f$ and $g$ are Boolean valued functions, i.e., the range of the functions is $\{\pm 1\}$. For real-valued functions this equivalence is not meaningful, since Hamming distance is not defined. This already highlights a key obstacle in relating their setting to ours.
>
> The main difficulty is that the SOSA'20 tester estimates distance to the class of $s$-Fourier-sparse real-valued functions under the Euclidean metric. In general, a function that is $\ell_2$-close to an $s$-Fourier-sparse real-valued function need not be Hamming-close to any $s$-Fourier-sparse Boolean function, unless the $\ell_2$ distance is very very small e.g. $o(\frac{1}{s^2})$. Taking signs does not resolve this, as the sign operation can destroy Fourier sparsity. To make the distinction explicit, let $\mathcal{F}(s)$
> denote the class of real-valued functions with at most $s$ nonzero Fourier coefficients. The tester of Yaroslavtsev--Zhou estimates
> $$
> \min_{g \in \mathcal{F}(s)} \|f - g\|_2^2,
> $$
> which effectively corresponds to approximating $f$ by truncating it to its top $s$ Fourier coefficients. However, the resulting function $g$ is not required to be Boolean. It is even hard to argue existence of nearby $s$-Fourier sparse Boolean function. This is why real-valued $s$-term $\ell_2$-approximations do not translate into guarantees for Boolean sparsity under Hamming distance. Thus, their tester does not address our problem: deciding whether a Boolean function is exactly $s$-Fourier-sparse, or Hamming-far from every Boolean function with at most $s$ nonzero Fourier coefficients. Hence, the upper bound in our work is not subsumed by the SOSA'20 result; the two settings are fundamentally incomparable in the way required for property testing over the Boolean class.
>
>
>
>
> We will clarify this distinction more explicitly in the revised version.

---

> > ### Author Response · Authors · 2025-11-27
> >
> > Dear Reviewer,
> >
> > We hope you have had the chance to go through our responses. If you have any further questions or require additional clarification, please let us know. we would be happy to address them to the best of our ability.
> >
> > Thank you once again for reviewing our work.

---

> > > ### Comment · Reviewer_x8Ha · 2025-11-28
> > > **Response to authors**
> > >
> > > I am satisfied with the clarification. I would be happy to see this discussion included the paper. I am happy to make my rating positive.

---

### Official Review · Reviewer_YARm · 2025-11-06

**Soundness:** 3
**Presentation:** 3
**Contribution:** 3
**Rating:** 8
**Confidence:** 3

**Summary:**

This paper studies testing the sparsity of Fourier coefficients of boolean functions under Hamming distance. The authors proves an upper bound of $\tilde{O}(s^{4})$ and a lower bound of $\Omega{s}$, significantly improving previous bounds.

**Strengths:**

1. Testing Fourier coefficients of boolean functions is a fundamental problem in theretical compter science. The authors significantly improve existing upper and lower bounds on testing sparsity under Hamming distance.
2. The techniques are interesting. It is inspired by prior works on testing junta functions, but requires non-trivial and novel design in the algorithms to work for testing sparsity in Fourier coefficients.

**Weaknesses:**

1. A recent work [1] obtain nearly tight bound for testing sparsity of Fourier coeficients under L2/Euclidean distance. While the distance metric is different, it is necessary to discuss differences in the technical ideas. The authors mention that closesness in Hamming distance implies closeness in Euclidean distance, and therefore it is more difficult to design algorithm for the former case, but this also means that lower bound for Euclidean distance could imply a lower bound for Hamming distance.

[1] Arijit Ghosh, Manmatha Roy, Price of Parsimony: Complexity of Fourier Sparsity Testing, to appear in NeurIPS 2025, https://openreview.net/forum?id=7bCPXHq8xV

**Questions:**

My main questions are related to the recent work [1]
1. What are the key distinctions with the algorithm used in [1]?
2. Does the lower bound in [1] imply a meaningfule lower bound for the Hamming distance?

[1] Arijit Ghosh, Manmatha Roy, Price of Parsimony: Complexity of Fourier Sparsity Testing, to appear in NeurIPS 2025, https://openreview.net/forum?id=7bCPXHq8xV

---

> ### Author Response · Authors · 2025-11-21
>
> We thank the reviewer for their thoughtful and careful evaluation of our work.
>
> The works of Ghosh-Roy (NeurIPS 2025) and Yaroslavtsev-Zhou (SOSA 2020) analyzes the proximity to the class of $s$-Fourier-sparse real-valued functions with $\ell_2$-norm at most $1$, under the $\ell_2$-distance metric. Note that $s$-sparse real-valued function may not necessarily be a $s$-Fourier sparse Boolean function. Consequently, their tester is insufficient for our setting, whose goal is to decide whether a Boolean function is $s$-sparse or is Hamming-far from every $s$-sparse Boolean function. Indeed a Boolean function may have $s$ large Fourier coefficients together with a long tail of many extremely small coefficients. Such a function can be close in $\ell_2$-distance to an $s$-sparse real-valued function, while possibly being far from every Boolean $s$-sparse function in Hamming distance. The tester of the Ghosh-Roy type cannot distinguish these two cases reliably. Therefore, their result does not subsume our upper bound: the two settings are incomparable. The revised version will make these distinctions more explicit.
>
>
> At a very high level, the techniques of Ghosh-Roy rely on sketching via hashing paradigm to detect heavy Fourier mass of the given function. Their analysis is related to the heavy-hitters framework. In contrast, our approach follows a different path. We first exploit the fact that every $s$-sparse Boolean function has Fourier support contained in a low-dimensional affine subspace. We then attempt to construct an equivalent low-dimensional $s$-sparse Boolean function (up to an invertible linear transformation). The construction relies on an exact learning procedure inspired by $\ell_1$-minimization–based compressed sensing, and ultimately reduces to generating sufficiently many random samples to run this reconstruction algorithm. Conceptually, our method fits into the broader paradigm of testing via implicit learning.
>
>
> While our lower bound proof and that of Ghosh-Roy use Maiorana-McFarland functions and analyze their behavior under singular and non-singular transformations, the problem settings are fundamentally different. It is correct that a lower bound for testing membership in a class of real-valued functions with unit $\ell_2$ norm implies a lower bound for testing membership in any subclass, such as $\{\pm 1\}$-valued functions, but this implication holds only when the distance measure is the same in both settings. When the distance metrics differ, such an inference need not hold. Also note that there is no known reduction from testing $\ell_2$-closeness of real-valued functions to $s$-sparsity, to the problem of testing whether a Boolean function is (or, even close to) $s$-sparse or far from every $s$-sparse Boolean function in Hamming distance. Therefore, the lower bounds do not automatically transfer. We will highlight these aspects in the revised version.
>
>
> Finally, we emphasize that our work is particularly relevant for the exact learning of Fourier-sparse Boolean functions. Exact reconstruction methods such as $\ell_1$-minimization–based compressed sensing succeed only when the target is (or, at least very close to) $s$-sparse, so certifying exact sparsity is essential. Our tester provides this certification: it rejects every function that is $\Omega(1/s^2)$-far (in Hamming distance) from the class of $s$-Fourier-sparse Boolean functions, making it a natural preprocessing step for exact reconstruction. By contrast, Goldreich--Levin aka Kushilevitz--Mansour–style PAC algorithms rely only on $\ell_2$-approximate sparsity. After taking signs, their outputs are close in Hamming distance but need not remain Fourier-sparse.
>
> In summary, our problem setting, techniques, and applications differ substantially from these prior works.

---

> > ### Author Response · Authors · 2025-11-27
> >
> > Dear Reviewer,
> >
> > We hope you have had the chance to go through our responses. If you have any further questions or require additional clarification, please let us know. we would be happy to address them to the best of our ability.
> >
> > Thank you once again for reviewing our work.

---

### Official Review · Reviewer_Dn21 · 2025-11-08

**Soundness:** 4
**Presentation:** 3
**Contribution:** 3
**Rating:** 8
**Confidence:** 3

**Summary:**

The paper presents new algorithms for testing Fourier sparsity of a boolean function given query access to that function. A boolean function is $s$-Fourier sparse if it's Fourier representation has at most $s$ nonzero components. This assumption is satisfied by classes of simple functions such as low-depth decision trees. Fourier-sparse family of functions that has received extensive theoretical investigation because of connections to a wide range of combinatorial and algorithmic questions.

The main results improve the known upper and lower bounds for the query complexity of testing Fourier-sparseness: the upper bound drops from about $s^{10}$ to about $s^4$. The lower bound jumps from about $\sqrt{s}$ to about $s$.

The paper achieves these results by refining existing approaches and developing new ones based on the structure of Fourier-sparse functions.

**Strengths:**

* This paper provides significant improvements on a clean algorithmic problem.

* The techniques seem like a nice contribution to the literature. I did not check the proofs, but the overall writing and presentation seem like a good fit for a specialist audience. It was nice to see many recent insights from the complexity/learning literature applied to this problem (which hadn't been directly attacked since 2011 or so).

* Applications of this result are not discussed in detail, but the authors point out that testers for Fourier-sparseness could, in principle, be used to help select the parameters of a learning algorithm that is to be run on a dataset.

* Section 3.1 does a nice issue of explaining the basic outline of the algorithm.

**Weaknesses:**

* My main hesitations about the paper have to do with significance and fit for ICLR. Specifically:

   * Exact vs tolerant testing: the authors suggest in the abstract that their algorithm could be used to select the parameter $s$ for a downstream learning algorithm that learns s-Fourier sparse functions. However, such an application would presumably require a tolerant tester, and would only be useful when the additional powers of $s$ in the testing results are dominated by the dimension-dependent factors required for learning. It would be good, for an ICLR audience, see this path spelled out—or at least a clear-eyed case be made for why this line of theoretical work should interest the ICLR audience.

   * Hamming versus Euclidean distance (similar to the tolerant testing issue above): The paper explains why testing for Hamming distance may be harder than testing for Euclidean distance, but not why that difference really matters for downstream use.

* The presentation is really tailored towards experts. Section 3.1 was pretty readable (modulo some notational issues), but Section 3.2 delved into specific jargon (e.g. "nonadaptive parity decision trees") pretty quickly.

* Minor issue: the dimensions and notation for the discussion of $f$, $f^*$, $L$, $U$, and $R$ (lines 200–206) was confusing. I guess you meant that $f = f^* \circ L$? and $R$ goes from $r$ to $n$ dimensions, not the other way around? In any case, a bit more explanation ehre would help, as would explicitly matching the notation with that of Section 3.2.

**Questions:**

I would like to see the authors make a clear and hones) case for the significance of their work to the ICLR community. Why is this the right paper for the venue and the right venue for the paper? Of the many possible lines of basic theory work that are adjacent to ML, which ones fit ICLR? (I guess "all of them" is a fine answer, but maybe not one that would see wide support.)

On a more technical level: what are the implications of this work for tolerant testing, if any? For what settings of parameters do these algorithms improve on those that actually learn a nearby s-Fourier sparse function? (What are specific settings where these algorithms would be useful for parameter selection?) Note that even query-based learning of s-Fourier sparse functions is already a fairly stylized theory question, albeit a fundamental and beautiful one.

---

> ### Author Response · Authors · 2025-11-21
>
> We thank the reviewer for their thoughtful and thorough comments.
>
> 1. Testing Fourier sparsity under the Hamming metric: While the problem of testing Fourier sparsity under the Hamming metric is of independent theoretical interest, its primary algorithmic relevance arises in the exact learning of Fourier-sparse Boolean functions $f : \mathbb{F}_2^n \to { \pm 1}.$ Exact reconstruction methods, such as compressed sensing based on $\ell_1$-minimization, succeed only when the target function is exactly, or at least very close to, an $s$-Fourier-sparse Boolean function. In these settings, either exact sparsity or the Hamming distance to the closest $s$-Fourier-sparse function should be below $o(1/s)$. Thus, one must first certify Fourier sparsity to have any meaningful guarantee on the output. Our tester provides precisely this certification: it rejects every function that is $\Omega(1/s^2)$-far (in Hamming distance) from the class of $s$-Fourier-sparse Boolean functions. Moreover, any two distinct $s$-Fourier-sparse Boolean functions differ on an $\Omega(1/s)$ fraction of inputs. Taken together, these facts imply that our tester cleanly separates exactly (or at least very close to) $s$-sparse functions from all others, making it a natural preprocessing step prior to exact recovery algorithms. In contrast, Goldreich-Levin (aka Kushilevitz-Mansour) style PAC algorithms for Fourier-sparse functions require only approximate Fourier sparsity in the $\ell_2$ sense. However, these algorithms output a hypothesis that is merely close in $\ell_2$, even when the target function is an exactly Fourier-sparse Boolean function. While taking signs yields a hypothesis close in Hamming distance, this sign operation can destroy Fourier sparsity, and therefore such algorithms do not return an exactly sparse representation even when one exists.
>
> 2. Tolerant testing: The reviewer is correct that tolerant testing is strictly harder than exact testing. Our algorithm can be adapted to obtain a tolerant tester as well. The key observation is that the subroutine SubsetImplicitCosetSampler draws $\tilde{O}(s^2)$ uniform random examples and checks whether there exists an $s$-Fourier-sparse Boolean function supported on a subspace of dimension about $\sqrt{s}$ that agrees with all sampled points. This step can be made robust to achieve tolerance. Specifically, in distinguishing whether a function is $\varepsilon$-close to some $s$-sparse function or $2\varepsilon$-far from every such function, we may allow up to an $\varepsilon$-fraction of the sampled labels to be flipped, and run the reconstruction procedure repeatedly on these perturbed batches of samples. With a mild degradation in complexity parameters, this yields an $\varepsilon$-versus-$2\varepsilon$ tolerant tester. In this work, however, we chose to focus on the exact (non-tolerant) setting. We believe that in the Hamming-metric context this is the more natural notion, for the reasons discussed above. For $\ell_2$-based formulations, we agree with the reviewer that tolerant testing is the more appropriate notion to consider.
>
> 3. Relevance to ICLR community: Broadly, structures admitting sparse representations in suitable bases, and the task of efficiently learning such structures, form a central theme of computational learning theory, represented prominently in ICML, NeurIPS, ICLR, and AAAI. For Boolean functions over the Hamming cube, sparsity in the Fourier basis is one of the most naturally occurring structural properties, with many works studying the learnability of subclasses of such functions. Our work addresses a natural question in this direction: the feasibility of exact reconstruction of Fourier-sparse Boolean functions. Given that one of ICLR’s core areas is computational learning theory, our paper aligns naturally with the scope of the venue. In addition, the problem of learning Fourier-sparse Boolean functions has recently found applications in practical machine learning, including but not limited to auditing of ML models (Ajarra-Ghosh-Basu, AAAI 2024) and hyperparameter optimization in deep learning (Hazan-Klivans-Yuan, ICLR 2018). We believe that more applications will continue to arise in settings where structural information about an unknown function must be inferred from black-box access, and in such contexts, sparsity testing may serve as a useful primitive. Finally, our approach combines ideas such as affine-invariant sketching, coset-based sampling, and dimension reduction for Fourier-sparse functions, and introduces new algorithmic techniques that may generalize beyond the specific problem of Fourier sparsity. For these reasons, we believe our work will be of genuine interest to the ICLR community.
>
> Finally, in the revised version we will improve the exposition for a broader audience by including intuitive proof ideas before presenting the formal proofs of all key lemmas and also expand the preliminaries section to cover all essential definitions.

---

> > ### Author Response · Authors · 2025-11-27
> >
> > Dear Reviewer,
> >
> > We hope you have had the chance to go through our responses. If you have any further questions or require additional clarification, please let us know. we would be happy to address them to the best of our ability.
> >
> > Thank you once again for reviewing our work.

---

### Meta-Review · Area_Chair_btDq · 2025-12-18

**Summary:**

This paper studies testing whether a function has sparse Fourier coefficients or far from being so. Reviewers have found that the problem is fundamental in learning theory, and the technical contribution is solid: authors improved both upper and lower bounds of sparsity testing of Boolean functions.

**Reviewer Concerns:**

I believe reviewers' concerns are addressed.

**Reviewer Scores:**

Reviewers will likely increase scores.

---

### Decision · Program_Chairs · 2026-01-26

Accept (Poster)